# Cyanoacrylate Glue for Treating Chronic Saphenous Vein Insufficiency: A Retrospective Observational Single-Center Study

**DOI:** 10.3390/diagnostics13142313

**Published:** 2023-07-08

**Authors:** Nicolas Falvo, Amine Latreche, Olivier Chevallier, Frédérik Ledan, Maud Jandot, Héla Daoud, Léo Fréchier, Romaric Loffroy

**Affiliations:** Image-Guided Therapy Center, ICMUB Laboratory, UMR CNRS 6302, Department of Vascular and Interventional Radiology, François-Mitterrand University Hospital, 14 Rue Paul Gaffarel, BP 77908, 21079 Dijon, France; nicolas.falvo@chu-dijon.fr (N.F.); aminelatreche2@gmail.com (A.L.); olivier.chevallier@chu-dijon.fr (O.C.); frederik.ledan@chu-dijon.fr (F.L.); maud.jandot@chu-dijon.fr (M.J.); hela.daoud@chu-dijon.fr (H.D.); leo.frechier@gmail.com (L.F.)

**Keywords:** endovenous cyanoacrylate ablation, venous insufficiency, quality of life

## Abstract

Endovenous thermal methods are superseding surgical stripping for treating chronic superficial venous disease but require tumescent anesthesia and can cause heat-related nerve injuries. Endovenous cyanoacrylate ablation is a more recent technique that does not share these drawbacks. A retrospective observational study of consecutive adults managed with endovenous cyanoacrylate was conducted in 2018–2021 at a single university center. The follow-up was 18 months. We identified factors associated with target vein closure at 18 months and measured changes in quality of life using the generic 36-item Short-Form (SF-36) tool and the venous disease-specific VEINES-QOL/Sym questionnaire. Adverse events were collected. In the 55 study patients with 67 treated veins, the closure rate at 18 months was 94% (95% CI, 85–98%). Target vein diameter ≥9.5 mm had 81% sensitivity and 75% specificity for predicting recanalization. Quality-of-life scores improved significantly (*p* < 0.001 for both surveys). The only adverse event was a type IV allergic reaction to cyanoacrylate that was resolved with corticosteroid and histamine-antagonist therapy. Endovenous cyanoacrylate ablation was highly effective and is safe in experienced hands. Studies are warranted to determine whether changes in the protocol increase the closure rate in patients with target veins ≥9.5 mm in diameter.

## 1. Introduction

Chronic venous disease is defined in the VEIN-TERM consensus document as any chronic morphological or functional abnormality of the venous system manifested in symptoms and/or signs requiring investigation and/or care [1]. The prevalence in France of chronic venous disease with symptoms of sufficient severity to impair quality of life was 40% in women and 17% in men in a 2012 survey [2].

Surgical stripping was the reference standard treatment for chronic venous disease until the introduction of endovenous radiofrequency and laser ablation methods in the late 1990s.

Although minimally invasive, these methods require multiple injections for tumescent anesthesia, which often cause pain and bruising [3]. Thermal injury to the saphenous nerves may also occur.

Cyanoacrylate glues are liquid embolic agents that polymerize upon contact with blood, occluding the vessel. They have been widely used for many years to occlude cerebral venous and arterial malformations, cerebral aneurysms, the ovarian veins, and the spermatic veins [4]. Among them, the most used is n-butyl-2-cyanoacrylate (NBCA). Endovenous cyanoacrylate injection for lower-limb varicosity ablation was introduced recently. This method does not require tumescent anesthesia and carries no risk of heat-related nerve injury.

Numerous studies have demonstrated the efficacy and safety of endovenous cyanoacrylate ablation for the treatment of superficial venous insufficiency. A randomized controlled trial of surgical stripping vs. endovenous cyanoacrylate ablation found 100% target vein closure at 3 months in both groups, with similar quality-of-life improvements but significantly less pain and bruising in the cyanoacrylate group [5]. Another randomized controlled trial compared endovenous cyanoacrylate ablation to radiofrequency ablation [6]. The 3-month closure rate was not inferior with cyanoacrylate (99% vs. 96% with radiofrequency) and showed a trend toward superiority. Pain was similar with the two methods, and bruising less with NBCA.

The main complication of cyanoacrylate ablation is endovenous glue-induced thrombosis (EGIT), which can extend across the saphenofemoral and saphenopopliteal junctions, causing deep vein thrombosis with a risk of pulmonary embolism. Manual compression of the junctions with the ultrasound probe is performed to prevent these complications. A 2020 systematic literature review with 1981 veins treated by cyanoacrylate, 445 by radiofrequency, and 484 by laser ablation showed a slightly but significantly higher 2-year closure rate with cyanoacrylate (93.7% vs. 90.9% and 91.5%, respectively) and a slightly lower rate of postablation deep vein thrombosis (0.5% vs. 0.9% and 0.6%, respectively) [7]. No cases of pulmonary embolism occurred in either group. Bruising and pain were less with cyanoacrylate. In a single-center retrospective study, cyanoacrylate and radiofrequency produced similar 12-month closure rates, but the procedure duration was significantly shorter (25 vs. 35 min, *p* < 0.001), and adverse events fewer (*p* < 0.05) with cyanoacrylate [8]. The nonthermal nature of cyanoacrylate ablation eliminates the risk of nerve injury. Moreover, the shorter procedure duration and absence of tumescent anesthesia may allow for treatment in the phlebology office, thereby saving time and lowering costs.

Some studies have looked for factors associated with repermeabilization of saphenous veins after treatment with endovascular glue. One of the main factors investigated was the diameter of the treated vessel, found with a cutoff of 8 mm in a 2017 study [9].

The objective of this single-center retrospective study was to evaluate the efficacy and safety of endovenous cyanoacrylate ablation to treat superficial venous insufficiency.

We will also investigate factors associated with repermeabilization, including the diameter of the treated vessel, to assess the benefits of increasing the dose of NBCA for larger veins (>8 mm).

## 2. Materials and Methods

### 2.1. Study Patients and Design

For this retrospective single-center study, we searched our institutional database to identify patients who underwent endovenous cyanoacrylate ablation to treat great or small saphenous vein insufficiency at the Dijon University Hospital (Dijon, France) between December 2018 and March 2021.

Institutional review board was not required for this study due to its retrospective nature, but our ethics committee approved this study. For this single-center cohort study, the requirement for patient informed consent was waived by the ethics committee in compliance with the French law on retrospective studies of deidentified health data.

The inclusion criteria were the age of 18 years or older and superficial insufficiency of the great or small saphenous vein demonstrated by Doppler ultrasound with the compression-release maneuver and/or Valsalva maneuver in the standing position. All targeted veins for treatment needed to demonstrate at least 0.5 s of reflux with a diameter of at least 3 mm.

We did not include patients who failed to attend all follow-up visits over the first 18 months. Patients who did not have the opportunity to complete the questionnaires at the 18-month visit were contacted later by phone to complete it. There was no deliberate selection bias by the authors to only perform the operation with favorable lesions.

### 2.2. Data Collection

The study data were collected by review of the medical charts, including the preprocedural and postprocedural ultrasound reports and procedural reports. We used standardized forms to record the clinical and ultrasound data collected before the procedure, including the clinical, etiological, anatomical, and pathophysiological (CEAP) class. The same data were collected for the follow-up visits seven days then one, three, six, and 18 months after the procedure. At each visit, bruising at the access site was sought by physical examination and vein recanalization and deep vein thrombosis by ultrasound. At baseline and at the 18-month visit, each patient completed two quality-of-life questionnaires, the 36-item Short-Form survey (SF-36 and the VEINES-QOL/Sym questionnaire with help from the physician if needed. The VEINES-QoL/Sym questionnaire specifically assesses the quality of life and symptoms of a range of conditions in chronic venous disorders of the leg [10]. The SF-36 is a generic QoL tool measuring health across three dimensions and including eight separate scales: functional status (physical functioning, social functioning, role limitations attributed to physical problems, and role limitations attributed to emotional problems), wellbeing (including mental health, energy and fatigue, and pain) and overall evaluation of health (including general health perception) [11]. Adverse events were also collected: major adverse events were defined as postoperative complications requiring admission or surgery and minor adverse events as postoperative complications requiring only medical treatment or monitoring (e.g., bruising, skin disorders, pain, puncture-site infection). For patients for whom the 18-month questionnaire was not available, a telephone contact was established, and the two quality-of-life questionnaires were completed.

The ultrasound scanners used for this study were the Aixplorer^®^ MultiWave (Supersonic Imagine, Aix-en-Provence, France) and the Aplio i800 (Toshiba, Minato City, Tokyo, Japan). All ultrasound scans and endovenous glue ablations were performed by the same physician (N.F.), who had six years of experience in endovenous cyanoacrylate ablation.

### 2.3. Outcome Measures

The primary outcome was the 18-month closure rate assessed by ultrasound in all patients at all stages. Closure was defined as complete obstruction of the vascular lumen along the entire length of the treated segment. The diameter and depth of the saphenous veins were measured at three levels: proximal, medial, and distal. Vessel depth was defined as the distance from the skin surface to the proximal vessel wall under minimal compression. Mean diameter and depth were defined as the average of the three measurements made.

Reflux was defined as retrograde gravitational flow for more than 0.5 s in a superficial vein having a diameter greater than 3 mm.

The secondary outcomes were the SF-36 and the VEINES-QOL/Sym scores and the rates of adverse events.

### 2.4. Procedure

All procedures were performed according to a rigorous standardized protocol, using the VenaSeal™ System (Medtronic, Dublin, Ireland), which injects aliquots of NBCA along the vein (Figure 1). This study drug is a nonthermal endovenous treatment that does not require tumescent anesthesia, allowing the procedure to be performed in the office and ultimately saving time.

Moreover, this is the only technique that eliminates the risk of nerve damage during treatment of the saphenous vein found in some thermal procedures.

Local anesthesia of the puncture site was the first step. The saphenous vein was punctured under the Doppler ultrasound guidance, and a 7-Fr introducer was inserted. A dilator was advanced over a guide to the saphenofemoral or saphenopopliteal junction, depending on the target vein. The dilator was then replaced by the device catheter, whose tip was placed 5 cm from the saphenofemoral junction or 3 cm for the saphenopopliteal junction.

The dispenser gun was designed to deliver 0.1 mL of NBCA with each trigger pull.

The gun was triggered once to deliver the first NBCA aliquot. The first administration of the product consisted of a double injection (about 0.2 mL) followed by a local compression of the junction for 3 min.

The second aliquot of 0.1 mL was delivered after pulling the catheter back 1 cm. After a 3-cm pullback, compression was applied to the vein for 3 min, using the ultrasound probe. The catheter was again withdrawn by 3 cm, and an aliquot of 0.1 mL was administered, followed by compression for 30 s. This maneuver was repeated every 3 cm along the entire length of the vein, regardless of its length and the type of vein treated (Figure 2). However, the last aliquot was administered 3 cm from the catheter access site. The catheter was removed, and complete occlusion of the vein was verified by ultrasound scanning of the entire length of the vein and to rule out spillage of glue into the deep venous system Manual compression was applied during the scan to check the absence of deep vein thrombosis. Additional phlebectomy under local anesthesia or other additional procedures were performed as deemed necessary by the operator.

After the procedure, patients were kept under observation in the postanesthesia care unit for approximatively 1 h to ensure the early detection of immediate complications such as access-site bleeding. Patients were told they could resume their usual activities starting 1 h after the procedure and were prescribed Grade 2 compression stockings and preventive-dose anticoagulation for the first week. Compliance with the wearing of compression stockings was optimal due to the instructions given by the practitioner during the preoperative ultrasound, and immediately after the procedure.

Moreover, the realization of these procedures by cyanoacrylate is not carried out during the summer period in our department to reduce to a minimum this lack of compliance.

### 2.5. Statistical Analysis

Standard statistical summaries of patient characteristics were performed.

Categorical variables were described as numbers (%), and continuous variables as mean ± SD and median ± IQR. The normality of the distribution was checked with the Shapiro–Wilk test and the histogram.

Comparisons were with the chi-square test or Fisher’s exact test for categorical variables and with Student’s *t* test or the Mann–Whitney test for continuous variables.

A univariate analysis was carried out to look for associations linking several factors to target vein closure at month 18. The tested factors were age, sex, and body mass index (BMI); vein depth and diameter; saphenous vein targeted (great vs. small); whether additional phlebectomy was performed; and whether preventive anticoagulation was given. The factors yielding *p*-values smaller than 0.20 by univariate analysis were entered into a multivariate logistic regression model. The following four clinically relevant variables were forced into the model: vein diameter, vein depth, BMI, and additional phlebectomy. For each variable, the odds ratio (OR) with its 95% confidence interval (95% CI) was computed.

In the event that we would identify one or more variables independently associated with the primary outcome (closure at month 18), we planned to determine the best cutoff for predicting recanalization by month 18, by computing Youden’s index and plotting the receiver operating characteristics (ROC) curve.

The statistical analysis was carried out using R (version 4.2.1). The *p*-values lower than 0.05 were taken to indicate statistically significant differences.

As the data were collected retrospectively, only patients with all parameters presents were included in this study, so there were no missing data. For some patients for whom only the 18-month questionnaire had not been completed, a telephone interview was used to retrieve these data.

## 3. Results

### 3.1. Patients and Target Veins

We included 55 patients with 67 treated veins. Table 1 reports their main features. Subject comorbidities included hypertension (*n* = 23), abnormal lipids (*n* = 17), diabetes (*n* = 8), and drug allergy (*n* = 2).

All patients were seen by the practitioner during the different consultations to attest the target vein closure, and eight patients completed the quality-of-life questionnaires later during a telephone interview. No patients were lost to the follow-up.

One of the most well-accepted methods to evaluate the clinical signs of venous reflux disease is the clinical, etiological, anatomical, and pathophysiological (CEAP) system. The CEAP includes four categories of assessment: the clinical signs of the disease (C), the etiology or cause of the disease (E), the anatomy of the disease (A), and its underlying pathophysiology (P). Based on these assessments, the severity of the disease can be classified.

Stage ranges from C0 to C6 (C0: no visible or palpable varicose veins; C1: telangiectasia or reticular veins; C2: varicose veins; C3: edema; C4a: pigmentation or eczema; C4b: lipodermatosclerosis or atrophie blanche; C6: active venous ulcer).

The most common CEAP class before the procedure was 3 (ankle edema due to varicose veins). The reasons for starting preventive anticoagulation at the time of the procedure were atrial fibrillation and a history of pulmonary embolism without diagnosed deep vein thrombosis. VenaSeal was used out of current recommendations and off label in CEAP class 1 in five patients. Indeed, telangiectasia and reticular veins were the main clinical signs of the disease in those patients who were complaining enough to consider treatment. Glue was then chosen because of its safety and non-invasiveness.

Patients with atrial fibrillation are on preventive anticoagulant therapy for life, or until their condition is treated. Patients with pulmonary embolism without deep vein thrombosis are treated for 6 months. Both types of patients were on preventive anticoagulation at the time the procedure was scheduled.

Some patients had already benefited from a treatment for varicose veins on the other leg, either by radiofrequency or laser, and decided to have a cyanoacrylate treatment in agreement with the practitioner to avoid tumescent anesthesia in particular.

A total of 32 GSV and 35 SSV were treated. The mean size (mm) of the treated veins was 8.5± 2.4 (6–16).

Table 2 reports the diameter and depth of the target veins.

### 3.2. Outcome Measures

Immediately post-procedure and at the 48 h follow-up, all patients had complete occlusion of the treated vein segments as assessed by duplex ultrasound.

The closure rates were 100% at one month, 98.5% at 6 months, and 97.0% at 12 months (Figure 3). At 18 months, the closure rate was 94.0% (95% confidence interval, 84.7–98.0%).

### 3.3. Factors Associated with Target Vein Closure at Month 18

By the univariate analysis (Table 3), only vein diameter was significantly associated with closure at month 18: closure was less common for wider veins. Age, gender, type of vein treated, or BMI are not factors favoring the vessel repermeabilization thought.

The multivariate analysis (Table 4) also showed that only target vein diameter was significantly associated with closure at month 18, with wider veins being less often occluded.

The best vein-diameter cutoff for predicting vein closure at month 18 was 9.5 mm. Table 5 shows the diagnostic performance characteristics of the cutoff. Figure 4 is the ROC curve.

### 3.4. Quality of Life

Figure 5 summarize the SF-36 and QoL scores at baseline and 18 months visits. Both the total SF-36 score and the VEINES-QOL/Sym score improved significantly from baseline to month 18 (*p* < 0.001 for both comparisons). It should be noted that these questionnaires take into account all venous disease, including non-GSV disease and contralateral signs and symptoms.

### 3.5. Adverse Events

There were no life-threatening adverse events. No technical failures occurred. There were no major adverse events involving the target veins, and neither did any patient experience access site infection or substantial hematoma formation. No thromboembolic events (phlebitis or glue-induced thrombosis) were recorded throughout the 18-month follow-up.

The only adverse event was a supposed type IV delayed hypersensitivity reaction to cyanoacrylate in one patient one week after the procedure. This is a delayed hypersensitivity, a T-cell-mediated immune response that occurs within 24 to 48 h, in response to a recognized foreign antigen. Erythema, pain, edema, and pruritus developed in the area of the target vein, suggesting phlebitis. This single posttreatment complication is also called CHAIR (complex hypersensitivity and irritation reaction). An ultrasound scan found no evidence of thrombosis. Corticosteroid and antihistamine therapy were effective in ensuring rapid resolution of the symptoms. Allergology tests confirmed the diagnosis of hypersensitivity to NBCA. Real type IV delayed hypersensitivity is supposed to be more difficult to control by medication than glue-induced chemical inflammation since allergen is still present inside the vein. In such case, to achieve symptoms reduction, the surgical removal of polymerized glue from the body may be necessary, but it was not needed here.

## 4. Discussion

This is a single-center retrospective study of the use of cyanoacrylate glue in the treatment of chronic saphenous vein insufficiency.

The rate of vein closure at month 18 was very high. A larger target vein diameter was significantly associated with treatment failure: a value of 9.5 mm or more predicted patency at month 18 with 81% sensitivity and 75% specificity. The depth of the target vein was not significantly associated with the closure rate. There were no instances of paresthesia, deep vein thrombosis, or pulmonary embolism. Generic and venous disease-specific quality-of-life scores were significantly improved at month 18 compared to baseline.

Endovenous cyanoacrylate ablation was compared to surgical stripping in a randomized controlled trial with 63 patients in each group [5]. The 12-month closure rate was 100% in both groups, but the pain and bruising scores were significantly lower in the NBCA group. Minor adverse events occurred in nine patients in the NCBA group, and twenty in the surgical stripping group, with this last having the only major complication.

Endovenous cyanoacrylate ablation has advantages over heat-based ablation by radiofrequency or laser: the risk of thermal injury to nerves is eliminated, and tumescent anesthesia is not required. Tumescent anesthesia is used to protect the perivenous tissues from the heat induced by thermal treatments. However, injuries can be caused by its administration, which requires several percutaneous injections. The needle can damage the saphenous or sural nerves if the treatment is performed below the knee in the case of treatment of small saphenous veins. Perforations of the saphenous vein wall, the fascial compartment surrounding the saphenous vein, or shearing of reticular veins or subcutaneous arterioles can add to postprocedural pain and bruising [12]. In contrast, cyanoacrylate glue treatment is not thermal and did not result in any complications such as paresthesia, postprocedural pain, or bruising in our patient series. Needle injuries along the thigh and calf are avoided. Overall, this technique is a nonthermal, nontumescent, nonsclerosant therapy, with no need for tumescent anesthesia, no risk of thermal injury, rapid return to normal activities. Other standard cyanoacrylates are available on the market for endovascular purpose, especially for embolization, such as Glubran^®^2 (GEM, Viareggio, Italy) or MagicGlue^®^ (Balt, Montmorency, France). They are much more fluid and necessitate more caution for being used in the treatment of great saphenous vein incompetence because of the higher risk of potential migration. Furthermore, they need the use of a microcatheter because no specific kit exists for this purpose yet. However, their cheaper cost could bring them on the market in such a setting in the near future.

In a randomized controlled trial that used the month-3 target-vein closure rate as the primary outcome, cyanoacrylate ablation was not inferior to radiofrequency ablation (99% and 96%, respectively) [6]. Bruising was less with cyanoacrylate, whereas pain and all other adverse events were similar in the two groups. A 5-year extension of this trial showed occlusion rates of 91.4% with cyanoacrylate and 85.2% with radiofrequency, again demonstrating noninferiority [6]. Interestingly, no recanalization events occurred between 3 and 5 years, and quality-of-life measures remained improved throughout the 5-year follow-up. In a multicenter prospective study of 70 veins, the 1-year closure rate was 92.9%, and no patients experienced paresthesia or serious adverse events [13]. Another prospective study, with 140 veins, demonstrated occlusion in 97.9% of cases at 12 months, again with no serious adverse events [14]. Finally, in a smaller, single-center, prospective study, the 3-year closure rate was 94.7% [15]. Thus, the closure rates and safety profile in our study were consistent with previous reports.

In our study, the target vein diameter of 9.5 mm or more was associated with recanalization by month 18. A single-center retrospective study of 57 target veins in 29 patients found that recanalization by 12 months was significantly more common at or above 8 mm [16]. At the same center, the normal cyanoacrylate protocol was compared to a protocol involving one additional drop of cyanoacrylate for veins measuring 8 mm or more in diameter [9]. After a median follow-up of 24 months, the additional drop significantly improved closure rates which, however, remained significantly lower than those seen with narrower veins. These findings indicate a need for further studies aimed at tailoring the cyanoacrylate procedure to target vein diameter.

One of our patients experienced a hypersensitivity reaction to cyanoacrylate. Similar reactions have been reported for 4% to 25% of treated veins [17]. They must be distinguished from phlebitis, as the symptoms of erythema, pruritus, edema, and pain are shared by the two conditions [18]. No cases of anaphylactic shock have yet been reported in the various studies. Special allergy tests, such as patch tests by a dermatologist, are recommended if a patient is suspected of having an allergy to acrylate.

We recommended that patients wear compression stockings for one week. However, many studies did not use compression stockings. Adherence to this treatment is notoriously poor [19]. Whether prescribing compression stockings for a brief period after the procedure decreases the risk of venous thrombosis deserves evaluation. Indeed, according to manufacturer’s recommendation, there is no need to use compression with this technique. However, we may be wrongly considering that any intervention for treatment of leg varices may benefit from prophylactic compression stockings, although probably less needed with the use of glue than other techniques. A reminder of the importance of wearing compression stockings was given by the practitioner during the preoperative consultation as well as after the procedure to encourage compliance. In addition, the procedures were not performed during the summer months, as the heat is an important factor in nonadherence [19].

The limitations of our study include the retrospective design, which carries a risk of information bias. It included a relatively small sample size at a single center. Our study was observational, and endovenous cyanoacrylate ablation was not compared to other treatment methods. The recruitment at a single university hospital may have induced selection bias. A single experienced operator performed all the procedures. No standardized assessment tools were used to measure pain intensity or bruising. Finally, eight patients were unable to complete the 18-month questionnaire at the 18-month visit, and their data were collected using less reliable methods. It is of utmost importance to highlight the fact that a worldwide randomized trial, called the Spectrum Study, is currently going on, comparing surgery, thermal ablation, and glue therapy. This is the largest study on this topic, which should give us accurate short- and long-term outcomes in the near future.

Moreover, a formal cost–benefit analysis has not been performed, especially since the product remains expensive and is not reimbursed by the French social insurance system.

## 5. Conclusions

The endovenous cyanoacrylate was highly effective in achieving target vein closure at month 18. The target vein diameter greater than 9.5 mm was significantly associated with a lower closure rate. The adverse events were mild, with no instances of venous thrombosis or pulmonary embolism. The quality-of-life scores improved significantly after treatment. These findings support the use of endovenous cyanoacrylate ablation in patients with chronic superficial venous insufficiency. We are currently performing a multicenter randomized of VenaSeal therapy vs. endovenous thermal ablation or surgical stripping in this indication. This will be a large patient enrollment study (approximately 800 patients) involving at least 40 centers worldwide, which will also evaluate time to ulcer healing in the CEAP’s six subjects.

## Figures and Tables

**Figure 1 diagnostics-13-02313-f001:**
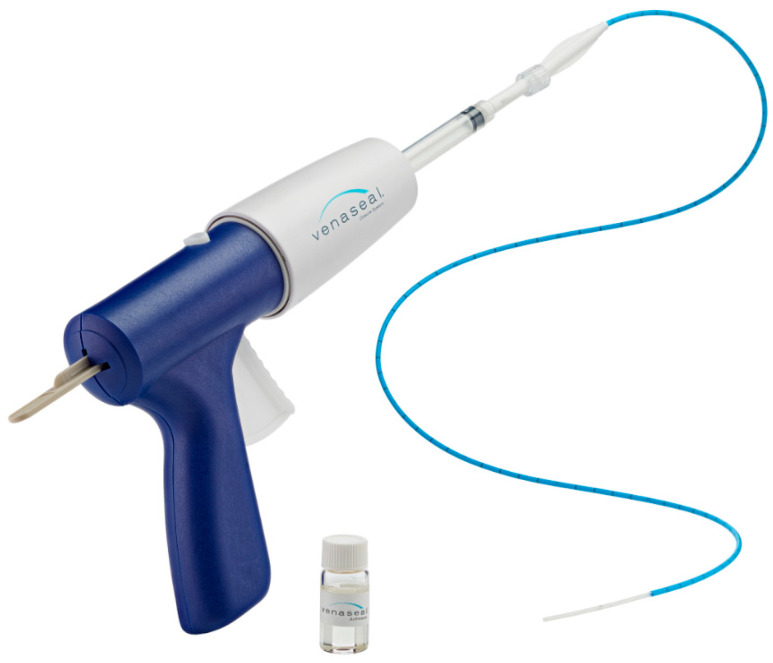
VenaSeal^TM^ closure system.

**Figure 2 diagnostics-13-02313-f002:**
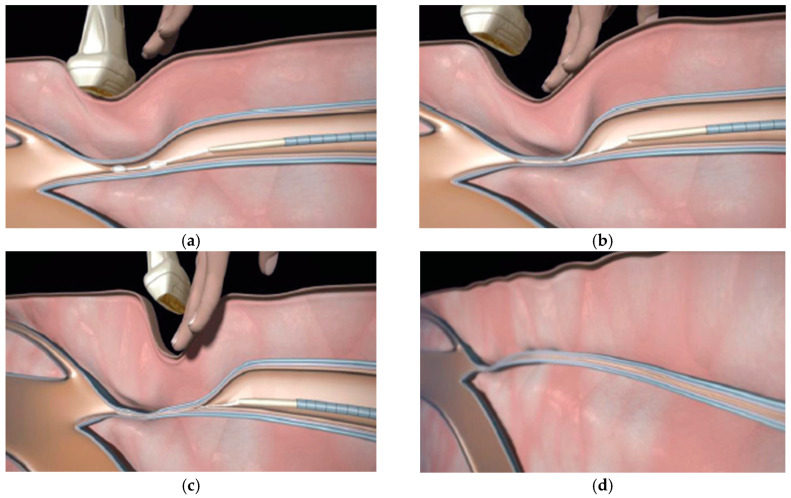
Typical procedural steps of VenaSeal^TM^ treatment. (**a**) Inject 0.10 cc adhesive into the vein, pullback for 1 cm, inject 0.10 cc, pullback for 3 cm. (**b**) Compress for 3 min. (**c**) Inject 0.10 cc, pullback for 3 cm, compress for 30 s. (**d**) Repeat process throughout the vein.

**Figure 3 diagnostics-13-02313-f003:**
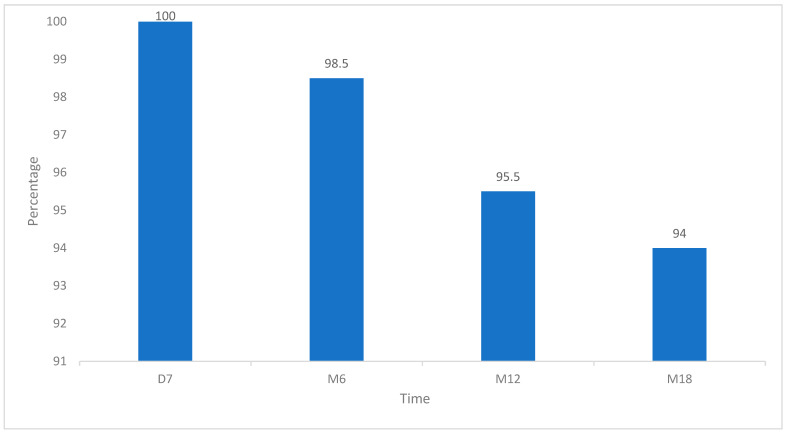
Proportion of patients with target vein closure at the four follow-up time points.

**Figure 4 diagnostics-13-02313-f004:**
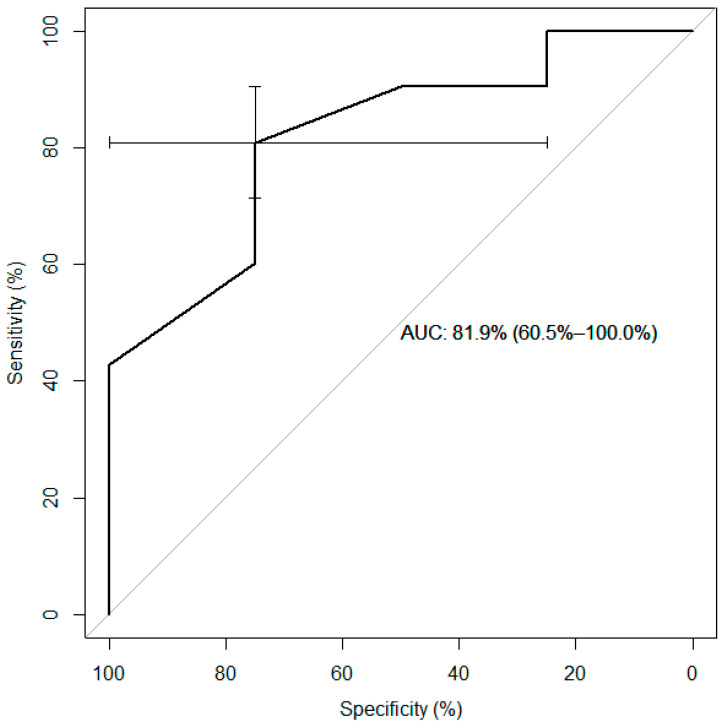
ROC curve of the performance of the 9.5 mm cutoff for vein diameter in predicting vein closure at 18 months.

**Figure 5 diagnostics-13-02313-f005:**
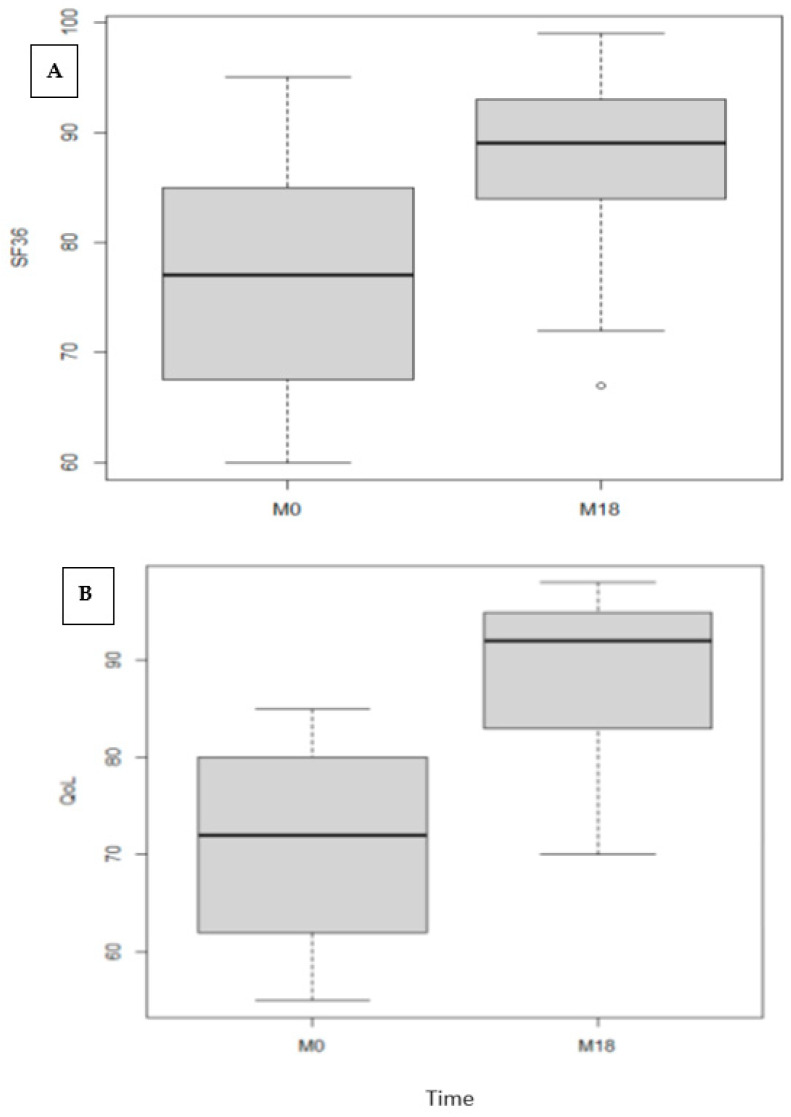
Box plots of the SF-36 score (**A**) and VEINES-QOL/Sym score (**B**) at baseline and 18 months after the procedure.

**Table 1 diagnostics-13-02313-t001:** Baseline characteristics of the patients and target veins.

Baseline Characteristics	Mean ± SD (Range)
Number of patients	55
Age, years, mean ± SD (range); median (IQR)	59.9 ± 15.5 (32–92); 60 (25)
Males/females (% females)	24/31 (56.4)
BMI, mean ± SD (range); median (IQR)	27.6 ± 5.9 (17.5–41.5); 27.3 (4.5)
CEAP class: C1/C2/C3/C4/C5/C6, *n* (%)	5 (7.5)/6 (9.0)/36 (53.7)/12 (17.9)/1 (1.5)/7 (10.4)
Patients with one treated vein, *n* (%)	45 (81)
Patients with two treated veins, *n* (%)	10 (19)
Great saphenous vein treated, *n* (%)	32 (48)
Small saphenous veins treated, *n* (%)	35 (52)
Patients with prior varicose vein treatment, *n* (%)	20 (30.8)
Patients given preventive anticoagulation, *n* (%)	12 (18)
Patients with additional phlebectomy, *n* (%)	12 (18)

BMI: body mass index; CEAP: clinical, etiological, anatomical, and pathophysiological classification.

**Table 2 diagnostics-13-02313-t002:** Diameter and depth of the target veins.

Veins	Number of Veins	Maximum Diameter (mm) Mean ± SD (Range)	Depth (cm) Mean ± SD (Range)
All	67	8.5 ± 2.4 (6–16)	11.2 ± 4.4 (5–22)
GSV	32	8 ± 2 (6–12)	13.8 ± 4.0 (8–22)
SSV	35	9 ± 3 (6–16)	8.2 ± 2.5 (5–15)

GSV: great saphenous vein; SSV: small saphenous vein.

**Table 3 diagnostics-13-02313-t003:** Univariate analysis of factors associated with target vein closure at month 18.

Variable	*p*-Value
Age	0.94
Sex	>0.99
Body mass index	0.31
Diameter of target vein	0.03
Depth of target vein	0.33
Target vein	0.62
Additional phlebectomy	0.56
Preventive anticoagulation	0.17

**Table 4 diagnostics-13-02313-t004:** Multivariate analysis of factors associated with target vein closure at month 18.

Variable	Odds Ratio (95% CI)	*p*-Value
Diameter of target vein	0.61 (0.39–0.94)	0.03
Depth of target vein	1.49 (0.87–2.56)	0.15
Body mass index	0.83 (0.63–1.11)	0.21
Additional phlebectomy	1.98 (0.05–8.05)	0.72
Preventive anticoagulation	0.086 (0.004–1.77)	0.11

**Table 5 diagnostics-13-02313-t005:** Diagnostic performance of the vein diameter cutoff of 9.5 mm.

	Value	95% CI
Sensitivity	0.8095	(0.6909–0.8975)
Specificity	0.7500	(0.1941–0.9937)
PPV	0.9808	(0.8974–0.9995)
NPV	0.2000	(0.0433–0.4809)
Accuracy	0.8060	(0.6911–0.8924)

PPV: positive predictive value; NPV: negative predictive value; 95% CI: 95% confidence interval.

## Data Availability

The data presented in this study are available on request from the corresponding author. The data are not publicly available due to restrictions aimed at protecting patient confidentiality.

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
