# Peer review of "Cyanoacrylate Glue for Treating Chronic Saphenous Vein Insufficiency: A Retrospective Observational Single-Center Study"

_diagnostics, 2023, doi:10.3390/diagnostics13142313_

Round 1

Reviewer 1 Report

Thanks to the authors for this nice original paper dealing with the treatment of great and small saphenous vein incompetence with a new cyanoacrylate. 

The results are promising, and the paper is well written. 

I have some comments : 

1. Could the authors please add some pictures of the device itself and the typical procedure ? 

2. Please describe a bit more the advantages of this specific glue for this purpose in the discussion section, and your opinion as compared to standard cyanoacrylates used for endovascular embolization. 

3. A word should be added into the last part of the discussion section about the worldwide ongoing Spectrum trial on this topic, in terms of perspectives. 

4. Not clear to me if all patients got systematic US controls during follow-up. Please describe a little bit more.

minor typewriting errors to improve

Author Response

Responses to Reviewer 1 comments

  1. Could the authors please add some pictures of the device itself and the typical procedure ?

Reply : Thank you very much for this comment. Pictures of the device and the procedural steps have been added in the « Procedure » section of the manuscript as requested. Figures in the text have been renumbered consequently.

  1. Please describe a bit more the advantages of this specific glue for this purpose in the discussion section, and your opinion as compared to standard cyanoacrylates used for endovascular embolization. 

Reply : Thank you very much for this comment. The advantages of this specific glue were already described in the discussion section. However, they have been completed and highlighted once again. We also gave our opinion in this section about the comparison with other standard glues on the market for embolization.

  1. A word should be added into the last part of the discussion section about the worldwide ongoing Spectrum trial on this topic, in terms of perspectives. 

Reply : Thank you very much for this comment. A sentence has been added in the last part of the discussion section about this trial.

  1. Not clear to me if all patients got systematic US controls during follow-up. Please describe a little bit more.

Reply : Thank you very much for this comment. It is already described in the data collection section and outcome measures section but it has been described a little bit more. Overall, indeed all patients got US follow-up.

Reviewer 2 Report

A clear and diligent report concerning the retrospective assessment of cyanoacrylate efficacy in a treatment of chronic venous disease with rather short observation period of 18 months. Despite an overall good impression, regrettably, this study is not novel and actually supports many other similar reports, concerning application of NBCA in CVD. Below are attached main comments to few minor flaws.

1. According to manufacturer’s recommendation and positioning in the market, the main benefit for patient from the use of NBCA there is no need to use tumescent anesthesia OR compression, either. However, in this study the CCL2 compression was recommended as a standard. It would be interesting for readers to know the Authors’ point of view regarding this discrepancy, not only the statement “as is” in current discussion (lines 340-347).

2. In the Table 1. five patients were classified as C1, according to CEAP. Authors should comment it, since, according to current recommendations and the Instruction for use for VenaSeal, this system is not intended to use in the treatment of telangiectasiae and reticular veins.

3. Authors have reported an adverse event, which was identified as type IV delayed hypersensitivity, with confirmed sensitization to NBCA. Interestingly, the administration of corticosteroids and antihistamines rapidly removed the symptoms (lines 277-285). However, here are two conditions which exclude each other. Although one could expect the fast improvement after medication would rather suggest a glue-induced chemical inflammation, than real type IV DH. The latter is difficult to control with allergen still present inside the vein. In such case, to achieve symptoms reduction, the surgical removal of polymerized glue from the body may be necessary (please, compare doi: 10.1016/j.jvsv.2021.06.014.).

4. Line 357 – Did Authors mean “French social insurance system”? Please, verify.

Author Response

Responses to Reviewer 2 comments

  1. According to manufacturer’s recommendation and positioning in the market, the main benefit for patient from the use of NBCA there is no need to use tumescent anesthesia OR compression, either. However, in this study the CCL2 compression was recommended as a standard. It would be interesting for readers to know the Authors’ point of view regarding this discrepancy, not only the statement “as is” in current discussion (lines 340-347).

Reply: Thank you very much for this comment. We full agree. However, we always prefer using compression stockings ffor all kind intervention on leg varices, whatever the technique, to avoid any risk of phlebitis. It has been explained in the discussion section more in details.

  1. In the Table 1. five patients were classified as C1, according to CEAP. Authors should comment it, since, according to current recommendations and the Instruction for use for VenaSeal, this system is not intended to use in the treatment of telangiectasiae and reticular veins.

Reply: Thank you very much for this comment. Indeed, the VenaSeal was used off label here in 5 patients. Telangiectasiae and reticular veins were the main clinical signs of the disease in those patients who were complaining enough to consider treatment. Glue was then chosen because of its safety and non-invasiveness. It has been explained in the text in the results section.

  1. Authors have reported an adverse event, which was identified as type IV delayed hypersensitivity, with confirmed sensitization to NBCA. Interestingly, the administration of corticosteroids and antihistamines rapidly removed the symptoms (lines 277-285). However, here are two conditions which exclude each other. Although one could expect the fast improvement after medication would rather suggest a glue-induced chemical inflammation, than real type IV DH. The latter is difficult to control with allergen still present inside the vein. In such case, to achieve symptoms reduction, the surgical removal of polymerized glue from the body may be necessary (please, compare doi: 10.1016/j.jvsv.2021.06.014.).

Reply: Thank you very much for this comment. We fully agree with this remark. However, the adverse event was considered as a type IV delayed hypersensitivity, responding well to medication, despite surprising. A paragraph in that section has been added to describe more in detail the differences in temrs of features and treatment.

  1. Line 357 – Did Authors mean “French social insurance system”? Please, verify.

Reply: Thank you very much for this comment. Indeed it is the French social insurance system. It has been corrected as requested.